# The state of primary healthcare centers in Saudi Arabia: A regional analysis for 2022

**Waleed Kattan** (ID) *

Department of Health Services and Hospitals Administration, Faculty of Economics and Administration, King Abdulaziz University, Jeddah, Saudi Arabia

* wmkattan@kau.edu.sa

## Abstract

### Objective

This study examines the 2022 distribution of primary healthcare centers across Saudi Arabia's 20 regions, focusing on disparities and healthcare accessibility.

### Methods

A quantitative analysis of the Ministry of Health's data was conducted. Primary healthcare centers distribution was evaluated by calculating the number of primary healthcare centers per 100,000 population across different administrative regions.

### Results

The study uncovered regional disparities, with the PHCs-per-100,000-people ratio showing significant variance. Regions like Riyadh had an above-average ratio with 7.5 PHCs-per-100,000-people, while Jeddah lagged behind with a concerning 3.2 PHCs-per-100,000-people despite being a populous city. The PHC-per-capita ratio declined to 6.6 in 2022 from 8.0 in 2017. The data also revealed that the Eastern Province showed an increase in the number of primary healthcare centers.

### Conclusion

Disparities in the distribution of primary healthcare centers in 2022 highlight a critical need for equitable healthcare access across Saudi Arabia. Many regions require increased primary healthcare center allocation to match population needs. The findings underscore the urgency of integrating these insights into policy frameworks to achieve the goals of Vision 2030, emphasizing the development of a sustainable and equitable healthcare system.

### Implications

Policymakers need to consider these disparities to guide the strategic placement of primary healthcare centers and ensure an equitable healthcare system. This study provides a basis for targeted policy interventions to improve healthcare equity and prepare the health system for future demographic and epidemiological transitions.

**Data Availability Statement:** All relevant data are within the paper.

**Funding:** The author(s) received no specific funding for this work.

**Competing interests:** The authors have declared that no competing interests exist.

## Introduction

### Importance of primary healthcare in Saudi Arabia

Primary healthcare centers (PHCs) play a crucial role in health systems by providing essential services to populations [1]. In Saudi Arabia, primary healthcare refers to basic health services delivered at the community level through PHCs, representing the first point of contact with the healthcare system [2, 3]. Studies show that primary care reduces illness and mortality by promoting prevention and accessibility of continuity care [4]. The multifaceted nature of primary care emphasizing prevention underscores its importance in comprehensive healthcare systems [5]. In recent years, Saudi Arabia has invested significantly in developing its primary healthcare infrastructure [6]. As the frontline for common health issues, PHCs will be integral to managing growing disease burdens [7].

### Regional distribution of PHCs in Saudi Arabia

The equitable regional distribution of PHCs is pivotal in ensuring accessible and comprehensive healthcare services in Saudi Arabia [8]. A thorough understanding of the current landscape of PHC distribution is instrumental in pinpointing areas that require enhancement, thereby addressing healthcare disparities effectively [9]. Countries that give precedence to primary healthcare by distributing resources equitably are more likely to accomplish their health and development objectives [10]. In Saudi Arabia, the way PHCs are spread across regions significantly influences the health outcomes and overall well-being of the population [11]. It is essential to ensure an equitable distribution of these centers to adequately respond to the growing population's needs [12]. Numerous studies underscore the necessity of continuously monitoring PHC distribution at the regional level to guarantee that healthcare services are uniformly accessible nationwide [2, 3, 13, 14].

### Saudi Arabia's healthcare transformation

In the context of Saudi Arabia's healthcare transformation, significant reforms have been implemented under the guidance of Vision 2030 and the National Transformation Program (NTP) 2020, with a particular focus on primary care enhancement [15]. Vision 2030 seeks to boost primary healthcare utilization [16], whereas NTP 2020 aims at fostering service integration and ensuring continuity of care [17]. Reflecting on the progress since the adoption of the Alma Ata Declaration in 1978, Asmri et al. [5] note the substantial expansion of the primary healthcare infrastructure, which has been instrumental in improving access to services via PHCs [18]. This infrastructure development is a key component in the broader strategy to optimize healthcare service provision across Saudi Arabia. A thorough analysis of the distribution of PHCs will facilitate more effective planning, ultimately contributing to the realization of the objectives set by Vision 2030 and NTP 2020 [4].

### Addressing regional disparities

The distribution of PHCs is intrinsically linked to regional disparities in healthcare access and outcomes [2]. Extensive research indicates that the way PHCs are distributed across regions significantly influences the accessibility of healthcare services in Saudi Arabia[2, 19, 20]. Achieving an equitable distribution of these centers is crucial not only for meeting the population's diverse needs but also for alleviating the burden of overcrowding in hospitals through the effective integration of primary care services [21, 22]. As the population continues to grow, strategically distributing healthcare resources becomes increasingly vital [23]. Regional disparities pose a significant challenge to the realization of healthcare goals, highlighting the need for

a comprehensive understanding of these disparities and the implementation of targeted interventions to address them [1, 24].

## Current study

This study examines regional PHC distribution across Saudi Arabia's 20 health regions. The overall PHC to population ratio and variation in provision across regions will be assessed. Findings provide policy-relevant insights regarding equitable service planning nationally under Vision 2030 strategies. Together with the reviewed literature, this study characterizes primary care development as foundational to Saudi Arabia's success in comprehensive, cost-effective, digitally-enhanced population healthcare nationwide.

This paper analyzes the regional distribution of PHCs in Saudi Arabia using 2022 Saudi Ministry of Health (MOH) data to characterize the progress of primary healthcare transformation. Comprehensive literature contextualizes primary care, strengthening centrality to Saudi national health system reforms targeting sustainable development goals through Vision 2030 strategies. Findings aim to guide efforts optimizing accessibility and advancing health equity objectives. This timely study contributes to understanding and informing Saudi Arabia's continued success in revolutionizing population health nationally through preventative, community-centered models empowering residents with equitable wellness nationwide.

## Materials and methods

This study compared the regional distribution of PHCs in Saudi Arabia for 2022 [25]. A secondary data analysis approach was employed using data from the MOH Statistical Yearbook. Data from the Saudi MOH Statistical Yearbook 2022, accessed on November 12, 2023, from the MOH website, was used for this study. This publicly accessible dataset provided aggregated and anonymized health-related statistics, with no identifiable information about individual participants, ensuring adherence to ethical standards for privacy and confidentiality in research.

The primary data used in this analysis was the number of PHCs available in each region. This information was extracted from the MOH Statistical Yearbook, which reported the total count of primary healthcare facilities at the end of the year. In addition to the PHC data, population estimates for each region were obtained from the demographic section of the MOH Statistical Yearbook.

The calculation of the number of PHCs per-100,000-people served as a measure for comparing the availability of primary healthcare services across regions. This metric considers the regions' varying population sizes, allowing for a fair and meaningful comparison of PHC distribution.

## Results

### Overview of hospital PHC capacity distribution (2017–2022)

Table 1 encapsulates a five-year overview of the dynamics between the population size and the availability of PHCs. Starting in 2017, the PHC count per-100,000-people began at 8.0 and

**Table 1. PHC availability vs. population growth in KSA, 2017–2022.**

| Year | Population | Total PHCs | Average PHC-per-100,000-people |
|------|-----------|-----------|-------------------------------|
| 2017 | 29,647,968 | 2,361 | 8.0 |
| 2018 | 30,196,281 | 2,390 | 7.9 |
| 2019 | 30,063,799 | 2,261 | 7.5 |
| 2020 | 31,552,510 | 2,257 | 7.2 |
| 2021 | 30,784,383 | 2,121 | 6.9 |
| 2022 | 32,175,224 | 2,120 | 6.6 |

showed a gradual decline to 6.6 by 2022. The population has expanded from approximately 29.65 million in 2017 to over 32.17 million in 2022, while the total number of PHCs has fluctuated slightly, resulting in a decreased PHC-per-capita ratio. This trend signals a relative decline in the per capita availability of primary healthcare resources against a steadily growing population.

The findings in the above table indicate that the total number of PHCs in Saudi Arabia remained relatively stable throughout the years, ranging from 2,120 to 2,390. However, the PHC-to-population ratio showed a gradual decline over the same period. In 2017, there were 8.0 PHCs-per-100,000-people, which decreased to 6.6 in 2022.

These results suggest that while the total number of PHCs did not significantly change, the growing population in Saudi Arabia has decreased the availability of PHCs per capita. This trend highlights the need for continued efforts to ensure adequate access to primary healthcare services across all regions in the country, considering the increasing population and their healthcare needs.

## Regional disparities in PHC allocation

In 2022, the total number of PHCs evaluated in this study amounted to 2,120 facilities, serving an aggregate population of 32,175,220 individuals. The national average stood at 6.6 PHCs-per-100,000-people, which provides a baseline for assessing the sufficiency and distribution of healthcare services across the country.

Table 2. shows that the distribution of PHCs is notably uneven when examined on a regional level. Al-Bahah, with a population of 465,779, boasts the highest ratio of PHCs-per-100,000-people at 20.0, significantly surpassing the national average. This indicates a more favorable healthcare landscape for residents of Al-Bahah, with easier access to primary healthcare facilities compared to other regions. Conversely, Jeddah, despite being one of the most populous cities with 4,623,940 inhabitants, has a markedly lower ratio of 2.1 PHCs-per-100,000-people, which is the lowest among all regions studied. This stark contrast elucidates a potential concern regarding healthcare accessibility in highly populated regions.

The data reveal a gradient in healthcare center distribution, with regions such as Bishah, Ha'il, and Qunfudah exhibiting PHC ratios of 16.6, 15.9, and 12.2, respectively, well above the national average. In contrast, more populous regions, including Riyadh, the Eastern region, and Makkah, present lower ratios of 4.8, 3.7, and 3.5, respectively, which could indicate a strain on healthcare resources and possible overcrowding in PHCs.

Further dissecting the data, Aseer, despite having a considerable population of 1,782,856, has managed to maintain a relatively high ratio of PHCs (11.8), suggesting a proactive approach to healthcare provisioning in this region. This starkly contrasts with the Eastern region, which shows a significantly lower ratio despite having a larger population size.

## Visualization of population distribution vs. PHC allocation

Fig 1 demonstrates the actual count of PHCs against the backdrop of the population size across various regions of the Kingdom of Saudi Arabia in 2022. Riyadh, being the most populous region, naturally has the highest number of PHCs. This could be seen as a direct response to the high demand for healthcare services in densely populated areas. However, the PHC count alone does not provide a complete understanding of healthcare accessibility. Regions like Al-Bahah, despite having a smaller population, have a comparatively high number of PHCs, which may indicate a strategic emphasis on healthcare distribution in less populous regions. The bar and line graph together highlight the raw numerical distribution of PHCs, allowing us

**Table 2. PHC distribution versus regional populations in KSA, 2022\*.**

| Rank | Region | Total Number of PHCs | Population | PHCs-per-100,000-people |
|---|---|---|---|---|
| 1 | **Al-Bahah** | 93 | 465,779 | **20.0** |
| 2 | **Bishah** | 63 | 380,622 | **16.6** |
| 3 | **Ha'il** | 109 | 685,571 | **15.9** |
| 4 | **Qunfudah** | 37 | 302,597 | **12.2** |
| 5 | **Al-Jouf** | 40 | 333,189 | **12.0** |
| 6 | **Najran** | 68 | 570,698 | **11.9** |
| 7 | **Aseer** | 211 | 1,782,856 | **11.8** |
| 8 | **Northern** | 41 | 358,989 | **11.4** |
| 9 | **Qaseem** | 152 | 1,397,187 | **10.9** |
| 10 | **Jazan** | 165 | 1,535,152 | **10.7** |
| 11 | **Tabouk** | 85 | 889,914 | **9.6** |
| 12 | **Qurayyat** | 16 | 165,804 | **9.6** |
| 13 | **Hafr Al-Baten** | 38 | 437,822 | **8.7** |
| 14 | **Ta'if** | 106 | 1,275,200 | **8.3** |
| 15 | **Medinah** | 146 | 2,105,376 | **6.9** |
| 16 | **Al -Ahsa** | 61 | 1,199,375 | **5.1** |
| 17 | **Riyadh** | 390 | 8,153,488 | **4.8** |
| 18 | **Eastern** | 120 | 3,202,857 | **3.7** |
| 19 | **Makkah** | 80 | 2,308,801 | **3.5** |
| 20 | **Jeddah** | 99 | 4,623,940 | **2.1** |
| | | **Total = 2,120** | **Total = 32,175,220** | **KSA average = 6.6** |

\*Table ranks regions by PHCs-per-100,000-people.

to identify regions where the number of healthcare facilities may exceed or not meet the needs of the population based solely on population counts.

In contrast to Figs 1 and 2 offers a more nuanced view of healthcare distribution by illustrating the PHC density per-100,000-people in relation to the population size in KSA for the year 2022. This figure provides a per capita analysis, giving insight into the relative availability

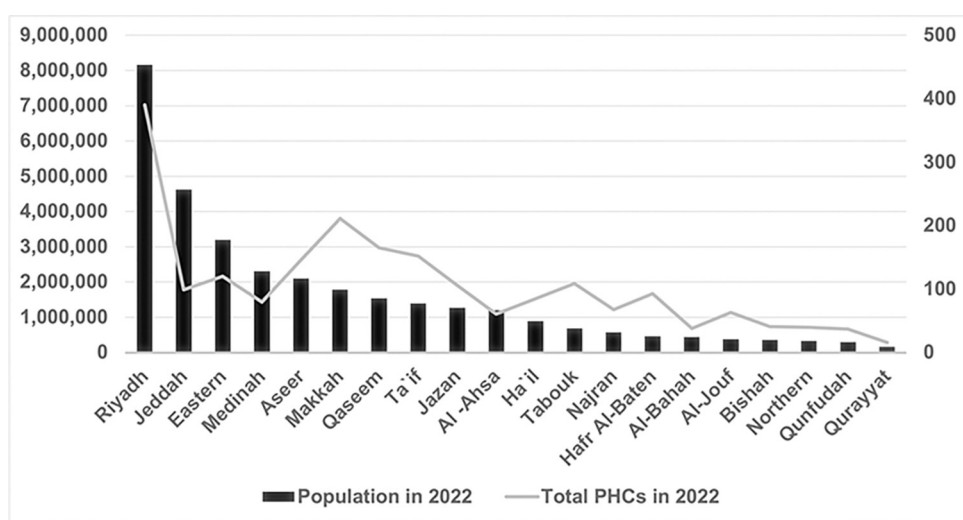

**Fig 1. Population versus PHC count by region in KSA for 2022.**

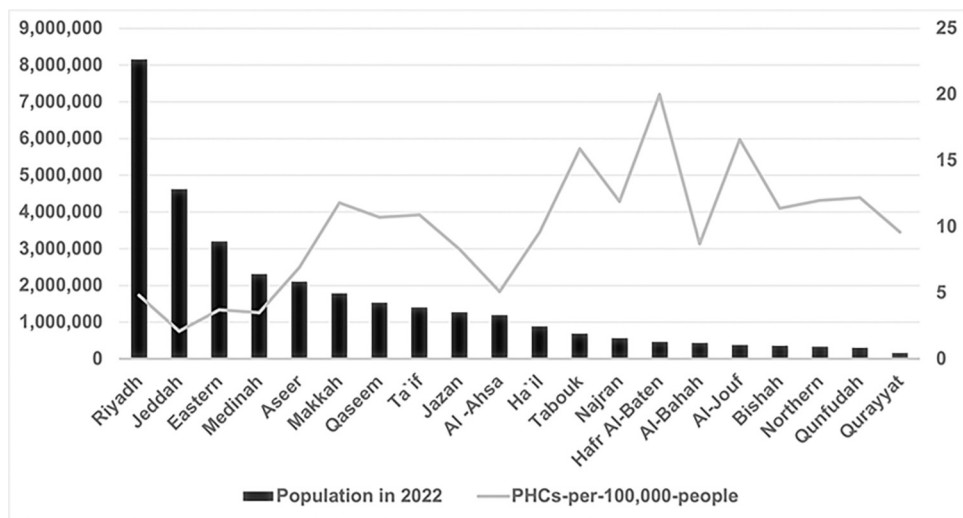

**Fig 2. Regional PHC density versus population in KSA, 2022.**

of healthcare facilities to the population in each region. It is evident that while Riyadh has the most PHCs, its PHC density per capita is relatively low compared to other regions, suggesting a potential underservice relative to its population size. Conversely, regions such as Al-Bahah have fewer PHCs but a higher density per capita, indicating a higher accessibility of healthcare services on a per-person basis. This figure underscores the importance of considering both the absolute and relative measures of healthcare availability to understand the actual landscape of healthcare accessibility across the regions.

Together, these two figures suggest that while some regions may appear well-served when considering the absolute number of healthcare facilities, a per capita perspective may reveal inequalities in healthcare distribution that could be addressed to improve the overall efficiency and equity of healthcare services in the Kingdom of Saudi Arabia.

## Discussion

### The regional divide in PHC distribution

The disparity in the distribution of PHCs between regions in Saudi Arabia is a significant concern. Some regions have better healthcare facilities than others, leading to unequal access to care and varying health outcomes [26, 27]. This regional divide is not unique to Saudi Arabia but is a common challenge in many countries, impacting the overall quality of life and health of the population [14, 27].

### Impact of population growth on PHC accessibility

The healthcare infrastructure in Saudi Arabia, particularly the availability of Primary Health Centers (PHCs), is being significantly influenced by the rapid population growth observed in recent years. In 2017, the PHC-per-capita ratio stood at 8.0 per 100,000 people, a figure that has since declined to 6.6 in 2022, signaling a potential gap in healthcare service provision [25]. This downward trend in PHC accessibility per capita is a matter of concern, indicating that the increase in PHC numbers has not been proportional to population growth. Moreover, this pattern highlights a critical oversight in the planning and development of healthcare facilities, which may not have adequately accounted for the demographic shifts and evolving healthcare demands of the population [7].

The declining PHC-per-capita ratio amidst Saudi Arabia's population increase necessitates a proactive reevaluation of PHC distribution strategies. Although the healthcare system was once adequate, it is evidently struggling to keep pace with current demographic and healthcare requirements. Strategic placement of new PHCs should be guided by comprehensive data analysis, considering geographic, demographic, and health trend variables to ensure equitable access. The diminishing ratio also risks increased strain on secondary and tertiary healthcare services, potentially leading to overcrowded hospitals and higher costs. Prompt, data-driven action is required to realign PHC infrastructure growth with population dynamics, securing a healthcare system that is both accessible and resilient for the future.

## Economic and policy factors influencing PHC distribution

The distribution pattern of PHCs can be attributed to various factors, including economic conditions, market dynamics, and government policies focusing on private sector development [18, 28]. However, this approach has resulted in an uneven distribution of PHCs, with some regions like Al-Bahah exceeding the national average of PHCs per capita, while populous cities like Jeddah have the lowest ratios [18, 23]. This indicates a need for a reassessment of allocation strategies to prioritize high-density areas with low PHC ratios.

The variance in the distribution of PHCs may be attributed to several factors, including but not limited to regional policies, geographical challenges, population growth rates, and the allocation of governmental healthcare funding. Regions with higher PHC-per-population ratios may benefit from strategic placement of healthcare facilities, effective regional health policies, or a combination of both. The lower-than-average ratios observed in some regions suggest an opportunity for policy interventions to enhance the accessibility and distribution of primary healthcare services.

## Investment in PHCs and healthcare outcomes

Investment in PHCs is essential for delivering cost-effective, patient-centered care [17, 29]. Establishing PHCs in underserved regions can enhance preventive care, reduce healthcare costs, and improve health outcomes [7, 12]. However, the current distribution suggests that the goals of Vision 2030, particularly those related to optimizing healthcare expenditure and transitioning to patient-centric care, are not fully realized [16].

## Challenges in access and quality of PHC services

Despite infrastructure development, issues such as the physical environment, waiting times, and location of centers continue to be barriers to access and use of PHCs [5, 27, 30]. The variation in services offered by PHCs in different regions suggests a need for a more standardized approach to healthcare provision across the country [21, 31].

## Enhancing primary healthcare in alignment with vision 2030

Vision 2030 is a strategic framework that focuses on revolutionizing the quality and accessibility of healthcare services through comprehensive sector reforms [15, 16]. Central to this vision is the Health Sector Transformation Program, which aims to restructure the healthcare system, thereby bolstering its capabilities and enhancing service provision [17]. A key focus of these reforms is improving quality, efficiency, and geographical coverage of healthcare services, ensuring universal access for all [16]. This initiative involves adopting value-based healthcare paradigms and aligning Saudi Arabia's strategies with global best practices to optimize the delivery of PHCs [16]. Furthermore, integrating digital innovations is seen as a pivotal step toward enhancing the efficiency and accessibility of healthcare services nationwide [32]. By

strengthening primary care through this comprehensive transformation, Vision 2030 lays the groundwork for achieving its broader objectives of fostering a vibrant society [16].

## Advancing healthcare through privatization and expansion

Under the Vision 2030 framework, a key strategy involves the privatization of 290 hospitals and 2,300 PHCs by 2030, aiming to elevate the private sector's contribution to the healthcare system from 25% to 35% [15]. This move towards public-private partnerships is expected to significantly enhance both infrastructure and service delivery within the healthcare sector [33]. Also, it is expected to significantly enhance both infrastructure and service delivery within the healthcare sector [7]. The Ministry of Health's plans underscore a growing commitment to developing the primary care sector, recognizing its critical role in the health system [18]. The flexibility offered by the private sector is seen as a complementary force in expanding healthcare facilities, particularly in underserved regions [18]. Moreover, collaborations among various stakeholders are crucial for addressing the challenges associated with resource distribution and enhancing the reach and quality of primary care nationwide [18].

## Advancing towards universal healthcare coverage

The implementation of insurance schemes is a strategic step towards ensuring universal access to primary healthcare services [27]. However, it is crucial to consider the implications of these insurance models on PHCs, particularly in terms of the pressures they may exert on the primary care system [34]. Adequate and equitable distribution of PHCs across regions is a crucial factor in the successful achievement of universal healthcare coverage [34]. This approach aligns with the principles outlined in the Astana Declaration, which advocates for the provision of comprehensive healthcare services throughout an individual's life course [10]. To realize this goal, it is essential to address the challenges posed by insurance schemes while simultaneously enhancing the capacity and reach of PHCs. This dual focus will help ensure that all individuals, regardless of their location or socioeconomic status, have access to the essential healthcare services they need.

## Digital health transformation

The deployment of digital health solutions has seen a significant acceleration in the wake of the COVID-19 pandemic, with telemedicine becoming an integral part of healthcare delivery nationwide [18, 35]. The advent of eHealth and digital innovations plays a crucial role in bridging the disparities between different areas, offering remote access to healthcare services [17, 36, 37]. As part of the Health Sector Transformation Program, there is a concerted effort to expand these digital services, recognizing their potential to enhance healthcare delivery [16]. Post-pandemic research underscores Saudi Arabia's notable strides in digital transformation, particularly in the realms of efficient disease surveillance and treatment [32] These digital solutions, optimized and promoted under Vision 2030, are pivotal in supporting equitable healthcare delivery across different regions. This digital shift not only improves accessibility but also ensures that healthcare services are more responsive and tailored to the population's evolving needs [4, 14]. By leveraging technology, Saudi Arabia is poised to create a more resilient and inclusive healthcare system, aligning with its broader goals of healthcare excellence and innovation [38, 39].

## Conclusion

In conclusion, this study has highlighted critical disparities in healthcare accessibility across different regions. Despite notable advancements in healthcare infrastructure, the findings

underscore a pressing need for equitable distribution of PHCs, particularly in urban areas and densely populated cities. The decreasing PHC-per-capita ratio amidst a growing population signals an urgent call for strategic planning and resource allocation. As Saudi Arabia strides towards achieving the ambitious goals of Vision 2030, it is imperative to address these disparities through informed policy interventions. The study's insights offer a valuable foundation for shaping a more robust, equitable, and sustainable primary healthcare system in the Kingdom that is responsive to its population's evolving demographic and health needs and sets a benchmark for future healthcare planning and implementation.

## Acknowledgments

I extend our sincere thanks to the Ministry of Health, Saudi Arabia, for making the essential data from their Statistical Yearbook publicly available, enabling this research.

## Author Contributions

**Conceptualization:** Waleed Kattan.

**Data curation:** Waleed Kattan.

**Formal analysis:** Waleed Kattan.

**Funding acquisition:** Waleed Kattan.

**Investigation:** Waleed Kattan.

**Methodology:** Waleed Kattan.

**Project administration:** Waleed Kattan.

**Resources:** Waleed Kattan.

**Software:** Waleed Kattan.

**Supervision:** Waleed Kattan.

**Validation:** Waleed Kattan.

**Visualization:** Waleed Kattan.

**Writing – original draft:** Waleed Kattan.

**Writing – review & editing:** Waleed Kattan.

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
