## [Decision Letter · Decision Letter 0]

16 Feb 2024

PONE-D-23-42418The State of Primary Healthcare Centers in Saudi Arabia: A Regional Analysis for 2022PLOS ONE

Dear Dr. Kattan,

Thank you for submitting your manuscript to PLOS ONE. After careful consideration, we feel that it has merit but does not fully meet PLOS ONE’s publication criteria as it currently stands. Therefore, we invite you to submit a revised version of the manuscript that addresses the points raised during the review process.

We look forward to receiving your revised manuscript.

Kind regards,

Masoud Behzadifar

Academic Editor

PLOS ONE

Journal Requirements:

2. We note that your Data Availability Statement is currently as follows: All relevant data are within the manuscript and its Supporting Information files

Reviewers' comments:

Reviewer's Responses to Questions

**Comments to the Author**

1. Is the manuscript technically sound, and do the data support the conclusions?

Reviewer #1: Yes

Reviewer #2: Yes

2. Has the statistical analysis been performed appropriately and rigorously? 

Reviewer #1: Yes

Reviewer #2: Yes

3. Have the authors made all data underlying the findings in their manuscript fully available?

Reviewer #1: Yes

Reviewer #2: Yes

4. Is the manuscript presented in an intelligible fashion and written in standard English?

Reviewer #1: Yes

Reviewer #2: Yes

5. Review Comments to the Author

Reviewer #1: The flow of the manuscript was excellent. The secondary data was suffice and the paper will get citations that will raise the impact factor of the Journal, due to the importance of the healthcare transformation which is part of the Kingdom's 2030 vision and strategic plan to improve the quality of healthcare services provided. Excellent job

Reviewer #2: Abstract- methods- spelling - primary-> Primary

Abstract- results- Urban regions like Riyadh - I recommend avoid using Urban / rural since Riyadh and Jeddah both Urban cities

Abstract- results- suggesting regional development efforts. - I dont see this to be included in results

Author waleed Waleed

Page 3- line 56 - burdens (7) -- add .

page 4 - line 65 - to cater - find a better word

PAge 4 - line 66- Numerous studies - in KSA ? or where? - it seems you are citing 13 ORCD - So I recommend some clarifications

Page 4 - line 69 - Saudi Arabia's Healthcare Transformation - the paragraph needs flow improvement

PAge 5 - line 89 - title Enhancing Healthcare in Alignment with Vision 2030 - add Primary

page 5 and 6 - 89 Enhancing Healthcare in Alignment with Vision 2030, 113 Advancing Towards Universal Healthcare Coverage, Digital Health Transformation - I believe to be moved to discussion

Page 8 Line 147- MOH - define - see line 157

Page 8 - 158 - which is a reliable and authoritative source for health-related statistics in the country - remove

159 - Ministry of Health (MOH) -  MOH

166- This data was crucial for assessing the - ?? remove?

170- standardized remove

159- A critical metric derived from the data was the

196 number of PHCs-per-100,000-people, facilitating a comparative analysis between regions with varying population densities ---- remove

Table 2 - PHCs - improve label - Number or Total number of PHCs, also Rank - *clarify footnote table

line 206- merge paragraph with above

206- (11.8 PHCs-per-100,000-people)  11.8

219-225 - move to discussion

233- initial- -remove

245-250 - I recommend improve the fig1, fig2 analysis

261 - 262- urban and rural - remove- this section as mentioned above

267- 267 Impact of Population Growth on PHC Accessibility - expand a little

290 Addressing Non-Communicable Diseases and Health Promotion - beyond the scope of the paper and the data - I recommend remove

296- this comprehensive study on the distribution of PHCs in Saudi Arabia in 2022 - this study

conclusion: These should focus on increasing the number of PHCs, enhancing the quality of care, and ensuring a standardized healthcare provision nationwide ( beyond the scope- replace)

Overall good findings and good study - er really need such papers on the Saudi system

6. PLOS authors have the option to publish the peer review history of their article (what does this mean?). If published, this will include your full peer review and any attached files.

Reviewer #1: **Yes: **Dr. Asaad A. Abduljawad

Reviewer #2: No

---

## [Author Response · Author response to Decision Letter 0]

21 Feb 2024

Dear Editor,

I have meticulously addressed the feedback provided by Reviewer #2. Changes, as suggested, have been implemented throughout the manuscript, including terminological precision in the abstract, clarification of the data's context, and restructuring for enhanced readability and focus. The revised manuscript now more accurately reflects the study's scope and strengthens its contribution to the literature on Saudi Arabia's healthcare transformation.

Warm regards,

Waleed

---

## [Decision Letter · Decision Letter 1]

26 Mar 2024

The State of Primary Healthcare Centers in Saudi Arabia: A Regional Analysis for 2022

PONE-D-23-42418R1

Dear Dr. Kattan,

We’re pleased to inform you that your manuscript has been judged scientifically suitable for publication and will be formally accepted for publication once it meets all outstanding technical requirements.

Kind regards,

Masoud Behzadifar

Academic Editor

PLOS ONE

Additional Editor Comments (optional):

Reviewers' comments:

Reviewer's Responses to Questions

**Comments to the Author**

1. If the authors have adequately addressed your comments raised in a previous round of review and you feel that this manuscript is now acceptable for publication, you may indicate that here to bypass the “Comments to the Author” section, enter your conflict of interest statement in the “Confidential to Editor” section, and submit your "Accept" recommendation.

Reviewer #1: All comments have been addressed

Reviewer #2: All comments have been addressed

2. Is the manuscript technically sound, and do the data support the conclusions?

Reviewer #1: Yes

Reviewer #2: Yes

3. Has the statistical analysis been performed appropriately and rigorously? 

Reviewer #1: Yes

Reviewer #2: Yes

4. Have the authors made all data underlying the findings in their manuscript fully available?

Reviewer #1: Yes

Reviewer #2: Yes

5. Is the manuscript presented in an intelligible fashion and written in standard English?

Reviewer #1: Yes

Reviewer #2: Yes

6. Review Comments to the Author

Reviewer #1: Everything is completed and the author has successfully addressed the reviewers requested items and the manuscript fulfilled the PLOS ONE requirements

Reviewer #2: I have reviewed the revised manuscript and am pleased to report that the authors have addressed all my previous comments comprehensively, enhancing both the methodological clarity and the depth of the analysis. The manuscript now presents a more robust and insightful contribution to the field. I commend the authors for their diligent revisions and believe the manuscript is ready for publication. Their efforts have significantly elevated the quality of the work, making it a valuable addition to the scientific community.

7. PLOS authors have the option to publish the peer review history of their article (what does this mean?). If published, this will include your full peer review and any attached files.

Reviewer #1: **Yes: **Asaad A. Abduljawad

Reviewer #2: No

---

## [Editor Report · Acceptance letter]

8 Apr 2024

PONE-D-23-42418R1 

PLOS ONE

Dear Dr. Kattan, 

I'm pleased to inform you that your manuscript has been deemed suitable for publication in PLOS ONE. Congratulations! Your manuscript is now being handed over to our production team.

Kind regards, 

on behalf of

Dr. Masoud Behzadifar 

Academic Editor

PLOS ONE